# An extensive spatiotemporal water quality dataset covering four decades (1980-2022) in China

Jingyu Lin[1,2], Peng Wang[3], Jinzhu Wang[4], Youping Zhou[5], Xudong Zhou[6], Pan Yang[1,2], Hao Zhang[7], Yanpeng Cai[1,2*], Zhifeng Yang[1,2]

[1]Guangdong Provincial Key Laboratory of Water Quality Improvement and Ecological Restoration for Watersheds, School of Ecology, Environment and Resources, Guangdong University of Technology, Guangzhou, 510006, China
[2]Southern Marine Science and Engineering Guangdong Laboratory (Guangzhou), Guangzhou, 511458, China
[3]Coastal and Ocean Management Institute, College of the Environment and Ecology, Xiamen University, Xiamen 361102, China
[4]School of Life and Environmental Sciences, Deakin University, Burwood, Vic, 3125, Australia
[5]Department of Ocean Science and Engineering, Southern University of Science and Technology (SUSTech), Shenzhen 518055, China
[6]Global Hydrological Prediction Center, Institute of Industrial Science, The University of Tokyo, Tokyo 153-8505, Japan
[7]CAS Key Laboratory of Tropical Marine Bio-Resources and Ecology, South China Sea Institute of Oceanology, Chinese Academy of Sciences, Guangzhou, 510301, China

*Correspondence to*: Yanpeng Cai (yanpeng.cai@gdut.edu.cn)

**Abstract** Water quality data represents a critical resource for the evaluation of aquatic ecosystems' well-being and the assurance of clean water sources for human populations. While the availability of water quality datasets is growing, the absence of a publicly accessible national water quality dataset for both inland and ocean in China has been notable. To address this issue, we utilized R and Python programming languages to collect, tidy, reorganize, curate, and compile three publicly available datasets, thereby creating an extensive spatiotemporal repository of surface water quality data for China. Distinguished as the most expansive, clean, and easily accessible water quality dataset in China by now, this repository comprised over 330,000 observations encompassing daily (3,588), weekly (217,751), and monthly (114,954) records of surface water quality spanning the period from 1980 to 2022. It spanned 18 distinct indicators, meticulously gathered at 2384 monitoring sites, which were further categorized as daily (244 sites), weekly (149 sites), and monthly (1,991 sites), ranging from inland locations to coastal and oceanic areas. This dataset will support studies relevant to the assessment, modelling, and projection of water quality, ocean biomass, and biodiversity in China, and therefore make substantial contributions to both national and global water resources management.

This water quality dataset and supplementary metadata are available for download on figshare repository at https://doi.org/10.6084/m9.figshare.22584742.v1 (Lin et al., 2023a).

# 1 Introduction

The implications of the 2030 Agenda for Sustainable Development necessitate the utilization of high-quality monitoring data for the purpose of gauging progress and facilitating evidence-based policymaking (Allen et al., 2021). Water, constituting the foundational pillar of sustainable development (UNESCO, 2019), bears a profound interconnection with numerous targets within the Sustainable Development Goals (SDGs), notably SDG 6 (Sadoff et al., 2020), which endeavours to ensure the universal availability and sustainable management of water and sanitation, and SDG 14, which focuses on the conservation and sustainable utilization of oceans, seas, and marine resources. With the campaign of ecological civilization and a series of marine policies (e.g., Maritime Power and Strategy, Chen et al., (2019)), China is committed to the preservation of water resources while simultaneously advancing resource management methodologies. To effectively accomplish the United Nations' SDGs and align with China's extensive policy frameworks, it is crucial to systematically compile water-related data across both inland and coastal/oceanic domains (Dai et al., 2022; Plagányi et al., 2023). Within the context of the Source-to-Sea (S2S) aquatic continuum, water quality data emerges as a pivotal factor in discerning pollution levels (Regnier et al., 2022). This information plays a critical role in the preservation of water resources and the provision of sanitation services (UNESCO, 2023).

Water quality refers to the selected physical, chemical, and biological characteristics of water that determine its suitability for a particular use (World Health Organization, 2017; Johnson et al., 1997). There are some key properties widely recognized for measuring water quality. In terms of physical characteristics, key considerations include the color, temperature (TEMP), sediment content, turbidity, electrical conductivity, and the concentration of Total Suspended Solids (TSSs) (Oteng-Peprah et al., 2018). Chemical constituents play a significant role in the determination of water quality. These encompass parameters such as the Potential of Hydrogen (pH), acidity levels, and indicators reflecting nutrient levels, including Ammonia Nitrogen (NH4N), Nitrite Nitrogen (NO2N), and Nitrate Nitrogen (NO3N), and various forms of phosphorus such as Dissolved Inorganic Phosphorus (DIP) and Total Phosphorus (TP). Additionally, the concentration of oxygen required for microorganisms to decompose organic matter is highly considered, which includes Biochemical Oxygen Demand (BOD), Chemical Oxygen Demand (COD), and Dissolved Oxygen (DO) (Hassan Omer, 2020). Biological indicators provide insights into the presence, condition, and abundance of various living organisms within water bodies, such as bacteria, algae, and pathogens. Overall, these indicators are crucial for assessing water quality and ensuring the health of aquatic ecosystems and human populations that rely on clean water sources.

Sustaining elevated water quality standards stands as an imperative requisite for the perpetuity of diverse spheres, encompassing natural ecosystems, public health, and socio-economic systems. Contaminants such as excessive nutrients that enter water bodies can have detrimental effects on the integrity, functioning, and biodiversity of both riverine and oceanic ecosystems which provide a habitat for a diverse array of flora and fauna (Morin and Artigas, 2023). For instance, the influx of pesticides into aquatic systems has been unequivocally associated with the diminishment of aquatic species and

perturbations in food chains (Stehle et al., 2015). Consequently, the unwavering adherence to stringent water quality standards emerges as an imperative measure for ameliorating the adversative consequences, thereby safeguarding fragile habitats, and preserving ecological equilibrium (Hering et al.,2015). Furthermore, the assurance of clean water represents a fundamental safeguard against the outbreak of waterborne maladies (Gleick and Palaniappan, 2010), with direct implications for the preservation of public health (Prüss-Ustün et al., 2014) and the concomitant mitigation of healthcare expenditures. Maladies

such as cholera, typhoid, and hepatitis find direct causation in the inadequacy of water quality (Leju Celestino Ladu et al., 2018). Lastly, impaired water quality can have severe economic consequences, including reduced agricultural productivity, increased costs of water treatment, and damage to tourism industries reliant on pristine water bodies (United Nations, 2018).

The recognition of the significance of the water quality to nature, society, food, and security has accelerated the arising and availability of local, national, and global water quality datasets. For example, local water quality datasets include the water

QUAlity, DIscharge and Catchment Attributes providing data for 1386 German catchments for the purpose of studying the species of nitrogen, phosphorus, and organic carbon (Ebeling et al., 2022), a set of water chemistry measurements including carbon species, dissolved nutrients, and major ions to describe the biogeochemical conditions of permafrost-affected in Arctic watersheds (Shogren et al., 2022), catchment-wide biogeochemical monitoring platform for capturing water temperature, pH, alkalinity, suspended solid, chlorophyll concentrations, and nutrient and cation data of the Thames basin in the United

Kingdom to promote drinking water resource management (Bowes et al., 2018). The Water Quality Portal (WQP) is comprising thousands of water quality variables encompassing physical conditions, chemical and bacteriological water analyses, chemical analyses of fish tissue, taxon abundance data, toxicity data, habitat assessment scores, and biological index scores, which was widely applied to lots of domains (e.g., to examine water clarity in lakes and reservoirs; Read et al., 2017). Aggregating five large water quality datasets, the Global River Water Quality Archive (GRQA) has significantly expanded

both the geographic and historical reach of existing water quality datasets by incorporating 42 parameters related to nutrient species, carbon content, sediment composition, and oxygen levels (Virro et al., 2021).

Despite significant advances in open data science for water quality research globally, Asia lags far behind other regions in this regard (Virro et al., 2021; Lin et al., 2023b). As the largest country in East Asia, China's water quality data are notably limited in the comprehensive global dataset, with a notable absence of data from coastal and oceanic regions. The publicly available

data consists of only 3595 daily observations in total from 244 sites, spanning from 1980 to 2009, as documented in GRQA. This is far from being adequate for water quality analysis and modelling. Additionally, the water data available from open data centres are stored in a user-unfriendly format that require significant additional efforts to make them credible, editable, and reusable. For example, monthly water quality data spanning from 2006 to 2022 are presented as reports with figures derived from statistical analysis, instead of providing more reliable monitoring data. Although some studies have employed national-

scale water quality data for assessment and modelling in China (Ma et al., 2020a; Ma et al., 2020b; Huang et al., 2021; Zhang et al., 2022), these datasets are not publicly available due to licensing restrictions and/or government-sanctions (Lin et al., 2023b). To date, there is no clean and publicly accessible national water quality dataset that covers the entirety of China.

Therefore, there is a pressing need to reorganize, curate, and manage the continuous, long-time series, standardized, well-organized, and consistent water quality datasets from inland to coastal/oceanic areas within China. These datasets stand as invaluable resources to support researchers and decision-makers (Van Vliet et al., 2023). They enable an in-depth examination of water quality status, encompassing the entire spectrum from riverine environments to the vast expanse of the oceans. Furthermore, they provide the means to model various dimensions of water quality indicators and forecast the ramifications of emergent water pollution phenomena (i.e., coastal eutrophication and oceanic harmful algal blooms due to additional nitrogen input from land and releases of radionuclides from inland redundant nuclear power plant accidents). It is also valuable to the effective management of water resources to support the United Nation Water Action Decade (2018-2028) and Ocean Decade (2021-2030; Folke et al., 2021). Our water quality dataset is thus initiated to meet the huge demand for Chinese water quality data, to boost national water data sharing, and to advance global water-related research and applications. It intends to collect non-sensitive and publicly available water quality data, to apply consistency to the formatting and curation, and to establish a standardized set of metadata for different water quality aspects.

## 2 Data and methods

### 2.1 Openly accessible data sources

The Chinese surface water quality dataset presented herein derived from three publicly accessible online data sources. Details of these original datasets were provided in Table 1.

**Table 1. Source datasets for compiling China water quality dataset.**

| Name | Data Sources | Timestep | Original observations (source/China) | Timeframe | Number of the parameters | Number of the sites (source/China) |
|---|---|---|---|---|---|---|
| Global daily water quality data | Global River Water Quality Archive (GRQA) | Daily | 17,000,000/3595 | 1898-2020 | 42 | 93,057/244 |
| National weekly water quality data | China National Environmental Monitoring Centre (CNEMC) | Weekly (7-day moving average) | 225,336/225,336 | 2007-2018 | 4 | 150/150 |
| National monthly water quality data | National Marine Environmental Monitoring Center (NMEMC) | monthly | 116,304/116,304 | 2017-2022 | 6 | 1991/1991 |

### 2.1.1 GRQA

As the most comprehensive water quality dataset, GRQA has incorporated inland water quality data from five existing sources, including the Canadian Environmental Sustainability Indicators program, Global Freshwater Quality Database, GLObal RIver Chemistry database, European Environment Agency, and USGS WQP for selected 42 water quality parameters (e.g., nutrients,

carbon, oxygen, and sediments; Read et al., 2017; Virro et al., 2021) with globally 93,057 sites in total spanning from 1898 to 2020 (Table 1).

### 2.1.2 CNEMC

As the most advanced and complete environmental data center, the China National Environmental Monitoring Centre (CNEMC) is an online information system managed by the agency of the China Ministry of Ecology and Environment. The
CNEMC was established in 1979 to monitor all environmental aspects (e.g., quality of air, water, soil), to provide publicly online data, to assess environmental impacts, and to report on the status of water environment for local and national governments. Water quality data available from this center included yearly water quality reports spanning from 2006 to 2022 (**http://www.cnemc.cn/jcbg/qgdbsszyb/index_6.shtml**), 7-day moving average (weekly) inland water quality data stored into individual WORD file or PDF file named by year with week number spanning from 2007 to 2018 (Table 1), real-time water
quality data for 11 indicators (TEMP, electrical conductivity, pH, DO, turbidity, $COD_{Mn}$, NH4H, TP, TN, Chlorophyll, and algal density) with a frequency of 4 hours (**https://szzdjc.cnemc.cn:8070/GJZ/Business/Publish/Main.html**), and real-time water quality data with a frequency of 1 month for 25 indicators (TEMP, electrical conductivity, pH, DO, turbidity, $COD_{Mn}$, BOD, $BOD_5$, NH4H, TP, TN, Fluorid, Cu, Zn, Se, As, Hg, Cd, Cr, Pb, Cyanide, Volatile Phenol, Total Petroleum Hydrocarbons (TPH), An-ionic Surfactant, and Sulfide) with data licensing and sharing restrictions. In this paper, we provided
the digital weekly water quality data which is publicly available.

This weekly water quality data was collected and constructed by following the standards from the *Environmental Quality standards for surface water (GB3838-2002)*. Water samples were automatically collected at six intervals throughout the day, with a sampling frequency of one sample every four hours (00:00-04:00, 04:00-08:00, 08:00-12:00, 12:00-16:00, 16:00-20:00, 20:00-24:00). The weekly water quality dataset was derived through the computation of daily averages encompassing Monday
through Sunday. This process yielded a single numerical value that served as a representative of a set of valid data samples. Specifically, a minimum of four data samples were aggregated to calculate the daily average, and five daily average data points were used to compute the weekly average.

### 2.1.3 NMEMC

Maintained by the China Ministry of Ecology and Environment since 2018, the National Marine Environmental Monitoring
Center (NMEMC) is an agency of a history of 60 years that specialized in marine ecological and environmental monitoring and protection. Monthly coastal/oceanic water quality data were accessible via **http://ep.nmemc.org.cn:8888/Water/** that were recorded from the year 2017 to 2023 and kept updated until now. Meanwhile weekly water quality reports of some important beaches along the coastal areas of China from 2019-2022 were available via **http://www.nmemc.org.cn/hjzl/hsycszzb/index.shtml** and annual average ocean ecological environment bulletins
**http://www.nmemc.org.cn/hjzl/sthjgb/**. Observation data were only available for monthly coastal/oceanic water quality data.

Guidelines in the *Specification for Offshore Environmental Monitoring (HJ 442-2008)* directed the methodologies, criteria, and quality assurance measures for monthly sampling of oceanic water quality. Employing Niskin and Go-Flo water samplers, samples were collected multiple times annually, typically during the months of April through December, as illustrated in Figure 1. The acquisition of this dataset entailed the collection of various quality control samples, including matrix spikes, blanks, parallels, and quality control check samples, which underwent meticulous collection and subsequent intra-laboratory comparison.

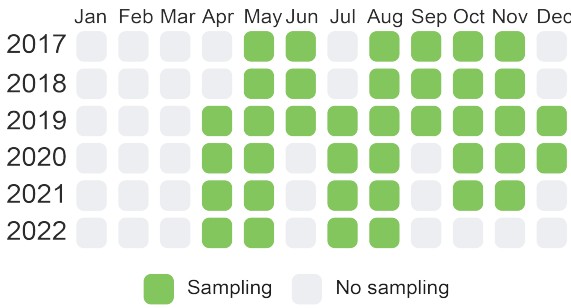

**Figure 1. Sampling frequency for oceanic water quality.**

## 2.2 Procedure for downloading and preprocessing source data

### 2.2.1 Data capturing

We extracted those sites located in China based on the geopolitical map after importing all coordinate data of the GRQA dataset into ArcGIS10.8. Afterwards, metadata information of countries/regions from GRQA were tidied and renamed for consistency. For instance, regions identified as "HK", "Macao", and "Taiwan" were renamed as "China". Therefore, we obtained daily water quality data in China from GRQA, which consisted of 244 sites for 15 selected water quality indicators (i.e., BOD, DO, COD, DIP, Dissolved Oxygen Saturation (DOSAT), NH4N, NO2N, NO3N, pH, Total Dissolved Phosphorus (TDP), TEMP, TP, TSSs, Dissolved Organic Carbon (DOC), and Total Organic Carbon (TOC)).

Weekly water quality data were tidied up from the reports collection derived from **http://www.cnemc.cn/sssj/szzdjczb/index.shtml**. To obtain all these files automatically, we inspected the elements of the webpage to locate the key nodes where *href* attribute specified the URL of the page the link goes for each report. Subsequently, a series of packages (i.e., *rvest*, *RSelenium*, *XML*, *purrr*) in R language were used to request remote URL and scrape the hyperlinks. A collection of hyperlinks was listed to download the original reports using *downloader* package. A total of 500 reports were identified, all of which were in WORD file format (i.e., DOC, DOCX, and PDF). These reports were originally designated with filenames that combined the year and the week number. Upon closer examination of the front-page summaries

in each report, it came to our attention that certain original report filenames exhibited inconsistencies with the actual content within. An illustrative example was the report labeled as "2010 - 1st week," which erroneously contained observations from the 37th week of the same year. A comparable situation arose with the reports for the 53rd week in the years 2011 and 2013, as revealed through an individual cross-referencing of filenames and report summaries. After the identification of these duplications, the affected files were expunged from the collection. Subsequently, a conversion process was undertaken to

transform each of these files into editable CSV files. These CSV files were then amalgamated into a unified worksheet file, comprising 11 columns. These columns encompassed a serial number, information on the watersheds (*MonitoringLocationDescriptionText*), the site name (*MonitoringLocationName*), the monitoring location type (e.g., river, lake, and reservoir; *MonitoringLocationType*), indicator values, the water quality index for the current week, the water quality index for the previous week, and descriptions on major pollutants. The dataset excluded the columns related to the water quality

index and major pollutants, as these columns mainly contained descriptive text intended to summarize information about water quality. The column containing the serial number was also excluded. The indicators featured in this dataset included DO, $COD_{Mn}$, NH4N, and pH.

We have collected the monthly coastal/oceanic water quality data from the NMEMC manually for the years 2017, 2018, 2019, 2020, 2021, and 2022. All data were stored as CSV files and were appended into a single worksheet file, which consisted of

14 columns (i.e., ocean's name (*MonitoringLocationDescriptionText*), province (*ProvinceName*), city (*CityCode*), code of the monitoring site (*Source_MonitoringLocationCode*), longitude (*LongitudeMeasure_WGS84*), latitude (*LongitudeMeasure_WGS84*), monitoring date (*MonitoringDate*), values of the indicators, water quality index for the current month). The column of the water quality index was removed. Indicators of the coastal/oceanic water quality data included COD, Dissolved Inorganic Nitrogen (DIN), DO, DIP, pH, and TPH.

**2.2.2 Coordinates of the monitoring sites**

Information of longitude and latitude is the fundamental information for identifying the location of a monitoring site. They were used to export spatial point data and were overlapped with other maps to obtain metadata information.

For daily water quality data, the longitude and latitude information were given by the GRQA dataset. Site location for weekly water quality data was coded as plain text of the administrative address, lacking geographic coordinates (i.e., longitude,

latitude). We first used geocoding API methods to find the address for a given place, thereby transforming the address into a corresponding geographic entity. Afterwards, we validated each of them by overlapping with the layers of watersheds and rivers according to the official maps obtained from the National Geomatics Center of China (http://www.ngcc.cn/ngcc/html/1/391/392/16114.html). All sites were confirmed to be located at the outlet of a river reach. As the geographic coordinates for the station labeled "Xuqiao" were unidentifiable from the provided information within the

original files, the data associated with this station were excluded from the dataset.

General information for the monthly coastal/oceanic water quality data was findable via the NMEMC. However, there were some information inconsistencies in longitude and latitude for the same station or place. For example, the station with code number FJD10003 was recorded with 120.57 E and 26.84 N in the year 2021 but with 120.58 E and 26.84 N in 2022. In addition, some stations with the same longitude and latitude may have different code numbers. Therefore, we first grouped them by code numbers and computed the average value of the longitude and latitude of that station to replace the initial value. Subsequently, we removed the column of the code number to avoid the same stations. Finally, we dropped the duplicated rows to get the unique stations.

All the transferred longitude and latitude information was merged into a single table and then imported into ArcGIS 10.8 as point shapefile in World Geodetic System 1984 (WGS84). After overlapping with the city-level administrative map and watersheds delineation map obtained from the National Geomatics Center of China, we derived other metadata information such as city, sub-watersheds (*MonitoringLocationTypeName*), etc. The code for the province (*ProvinceCode*) and city (*CityCode*) was referred to the China Area Code and Zip code of Version 2021.

### 2.2.3 Data cleaning and technical validation

We undertook a comprehensive standardization process across all the above-mentioned data providers. This harmonization encompassed the transformation of downloaded time series into a uniform file format, shifting from CSV files to R time series. Additionally, we ensured consistency in indicator selection, units, data structure, identification of missing values, and language.

Given the limited availability of indicators within the (sub)datasets, all of them were incorporated into our water quality dataset. This inclusive selection comprised both physical parameters (e.g., TEMP, TSSs) and chemical parameters (e.g., pH, BOD, COD, $COD_{Mn}$, DO, DOSAT, DIN, NH4N, NO2N, NO3N, TDP, DIP, TP, TPH, DOC, TOC). We adopted GRQA as a reference for indicator abbreviations, with the aim of facilitating international compatibility when appending to global datasets. It is noteworthy that, except for temperature (°C), pH, and DOSAT (%), the original unit of measurements for all indicators in the (sub)datasets was milligrams per liter (mg $L^{-1}$), and we retained this unit uniformity for consistency. Eight columns (i.e., *MonitoringLocationIdentifier*, *LongitudeMeasure_WGS84*, *LatitudeMeasure_WGS84*, *MonitoringDate* (with the format %d/%m/%y), *IndicatorsName*, *Value*, *Unit*, *SourceProvider*) were then included for structuring the full dataset. Column for *MonitoringLocationIdentifier* was created as an index to connect with the metadata file.

Some observations for different indicators were merged into a single column when converting the PDF file to editable files for weekly water quality data. Those columns were selected to be divided and tidied up into several columns via regular expression automatically and validation manually. Three additional columns were added to indicate the specific year (column *MonitoringYear*), week number (column *MonitoringWeek)*, and monitoring date (column *MonitoringDate*) for the weekly water quality data. The specific years and week numbers were subtracted from the filenames. The column of *MonitoringDate*

for that specific week was estimated using R according to the international standard ISO 8601 that Monday was considered the first day of a week. They were validated with the descriptive text on the cover of each report that was deleted later from the weekly water quality dataset. The column of *MonitoringDate* from ocean water quality data was assumed to occur on the first day of that month to keep consistency in the date format of other datasets.

In addition, duplicated rows were identified and removed by using distinct function in R based on the unique site, indicators, monitoring week/date, and values from the (sub)datasets that included 1776 site pairs from the weekly water quality dataset due to the file inconsistencies mentioned in 2.2.1. Negative values (with 7 observations) were omitted from the weekly water quality dataset. No duplicated rows and negative values were identified from the monthly water quality datasets. In cases where 7 sites provided two daily observations but lacked specific timestamp information from the GRQA, we substituted these records with the calculated average value of the two observations. Missing (e.g., noted as '-') and empty data were replaced with *NA*, and were omitted from the dataset. Values that falling below known detection limits were denoted as "< *DL*" within the monthly water quality datasets, which contain 3,490 data points. COD, DO, DIN, DIP, and TPH detection limits were 0.15 mg/L, 0.32 mg/L, 0.001 mg/L, 0.001 mg/L, and 0.001 mg/L, respectively. The descriptions in the stations that were originally in Chinese were replaced with Hanyu Pinyin.

## 2.3 Methods for quality assurance

Since data quality will generate bias and uncertainty for the results despite conducting imputation (Tiyasha et al., 2020), it was a necessary step to conduct data quality assurance to determine the shortcomings, errors, and issues of research results, and ensure robust study for different data users (Koelmans et al., 2019). In this paper, we used data availability and outliers for identifying quality assurance characteristics.

### 2.3.1 Availability

Data availability was characterized to assess the available records, both spatially and temporally. For each time series, we first counted the length of the records (*LengthofData*) to illustrate the general temporal coverage. Then, we assessed the data intensity, computed as the ratio between the length of the time series and the length of the time series without missing values. Furthermore, we used overall availability, longest availability, and continuity to measure the characteristics of availability following the methods from Crochemore et al. (2019).

### 2.3.2 Outliers detection and treatment

Outliers were detected by using the interquartile range (IQR) method. IQR is the range between the first (Q1) and third (Q3) quartile. Data points that fell below Q1-1.5×IQR and above Q3+1.5×IQR were considered outliers. Since it was difficult to determine whether an outlier is an error caused by faulty equipment or data entry errors or not, no observations were omitted from the original datasets.

## 3 Data Records

### 3.1 General information of metadata

All data were constructed in the form of CSV, while site information was provided with point shapefile (.shp) map (available for download at https://doi.org/10.6084/m9.figshare.22584742.v1). Referring to the inventory information of WQP, descriptions of the metadata for each time series of the water quality dataset were explained in Table 2.

**Table 2. Metadata information for water quality data**

| Field name | General introduction | Descriptions | Data type |
|---|---|---|---|
| ID | / | Identifier for each time series | Int |
| WaterDataType | Water data type within a broader aspect | "W2" stands for water quality data | String |
| MonitoringLocationIdentifier | Identifier for monitoring location | Identifiers for the stations | Int |
| MonitoringLocationDescriptionText | Given by the data source | | String |
| MonitoringLocationName | Given by the data source | Name of the station | String |
| MonitoringLocationType | Indicate the type of monitoring site | River, Lake, Reservoir, Ocean | String |
| MonitoringLocationTypeCode | Using code to indicate the type | River(R), Lake(L), Reservoir(V), Ocean(C) | Character |
| MonitoringLocationTypeName | Specific the name of that monitoring site | In which rivers, which lakes | String |
| Source_MonitoringLocationCode | Location code from the original datasets | | String |
| LongitudeMeasure_WGS84 | | | Float |
| LatitudeMeasure_WGS84 | | | Float |
| ProvinceName | The acronym of a specific province | | String |
| ProvinceCode | China area code and zip code | | Int |
| CityCode | China area code and zip code | | Int |
| IndicatorsName | | | String |
| IndicatorsUnit | | | String |
| ResolutionCode | Using numbers to identify the temporal resolution | | Int |
| ResolutionName | Temporal resolution | | String |
| CountryCode | | | Int |
| StartDate | | | Date |
| EndDate | | | Date |
| LengthofData | The count of observations in each time series | | Int |
| DataIntensity | Ratio between the length of the time series and the length of the time series without missing values | | Float |
| OverallAvailability | Length of the observation series, as a fraction of the dataset' longest period | Refers to Crochemore et al. (2019) | Float |
| LongestAvailability | Length of the longest observation series without gaps, as a fraction of the dataset' longest period | Refers to Crochemore et al. (2019) | Float |
| Continuity | Ratio between longest availability and overall availability | Refers to Crochemore et al. (2019) | Float |
| SourceProvider | Data source | | String |
| SourceProviderID | To separate the type of data source | Classified as authoritative and non-authoritative | String |

### 3.2 Spatial-temporal distribution of monitoring sites

After conducting cross-validation, it was observed that there was no spatial convergence among monitoring sites from different data sources (Figure 2). The dataset contained a large number of monitoring sites for the coastal and oceanic areas obtained

from NMENC (Figure 2). Most GRQA sites were located in tributaries, while the CNEMC provided most of the sites from the mainstream.

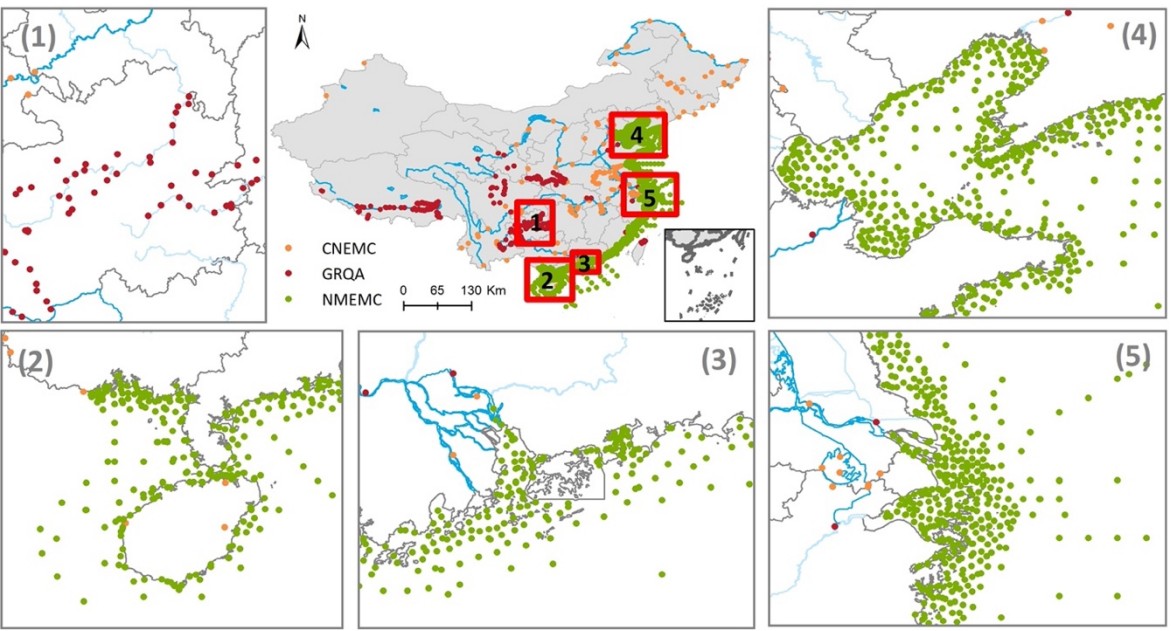

**Figure 2. Spatial distribution of water quality monitoring sites from different sources with drainages in China.**

Our dataset encompassed monitoring site records spanning from 1980 to 2022 (Figure 3). Number of sites for daily, weekly, and monthly observations were 244, 149, and 1991 respectively. Overall, the number of monitoring sites with records exhibited a slight increase before 2016, followed by a significant surge after 2016. Notably, GRQA predominantly contributes observations from monitoring sites prior to 2006, with an average of 133 observations obtained from approximately 13 sites per year, as illustrated in Figure 3a and Figure 3b. In contrast, CNEMC provides data from monitoring sites between 2007 and 2018, averaging around 126 sites per year, while NMEMC covers the period from 2017 to 2022 with an average of approximately 1249 sites per year. Despite CNEMC providing fewer monitoring sites, it consists of a comparable number of observations with an average of approximately 18,145 observations per year compared to NMEMC with an average of 19,159 observations. Comparatively, CNEMC and NMEMC datasets offer a greater number of records in comparison to GRQA. Temporal overlaps between various sources were identified on two occasions. The first instance transpired during the years 2007 to 2009, involving data from the GRQA and the CNEMC. The second temporal overlap was documented between CNEMC and NMEMC for the years 2017 to 2018.

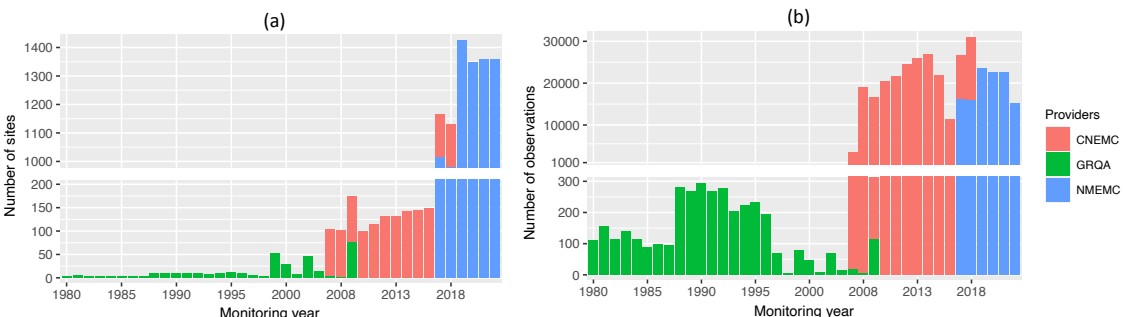

**Figure 3. Distribution of monitoring sites (a) and observations (b) from different sources over time.**

### 3.3 Characteristics of time series

The study has identified four distinct types of monitoring locations, comprising rivers, lakes, reservoirs, and coast/ocean (Table 3). The majority of the monitoring sites were located in the coast/ocean, with 1991 sites, followed by 365 sites in rivers that encompassed most of the indicators. Rivers from CNEMC demonstrated a considerable number of observations for $COD_{Mn}$, DO, NH4N, and pH indicators, while COD, DIN, DIP, DO, pH, and TPH indicators have the most observations in the ocean. Despite having fewer sites and observations for most indicators, rivers had a longer time series period compared to other types. Indicators of COD, DIP, and TPH exhibited some values that fell below the detection limits.

**Table 3. Stats for different types of the monitoring sites and indicators.**

| Location Type | Sites in total | Indicators' number | Indicators' name | Sites | Observations | Start date | End date | Below limits(n) | Outliers (%) | Sources(n) |
|---|---|---|---|---|---|---|---|---|---|---|
| Coast/Ocean | 1991 | 6 | COD | 1991 | 19,367 | 2017-05 | 2022-08 | 94 | 4.88 | NMEMC |
| | | | DIN | 1991 | 19,369 | 2017-05 | 2022-08 | / | 8.99 | NMEMC |
| | | | DIP | 1991 | 19,369 | 2017-05 | 2022-08 | 939 | 6.76 | NMEMC |
| | | | DO | 1991 | 18,143 | 2017-05 | 2022-08 | / | 2.78 | NMEMC |
| | | | pH | 1991 | 19,338 | 2017-05 | 2022-08 | / | 3.69 | NMEMC |
| | | | TPH | 1991 | 19,368 | 2017-05 | 2022-08 | 2453 | 2.88 | NMEMC |
| River | 366 | 15 | BOD | 10 | 432 | 1980-01-07 | 1997-11-27 | / | 6.71 | GRQA |
| | | | COD | 10 | 235 | 1988-01-03 | 1997-11-27 | / | 6.81 | GRQA |
| | | | $COD_{Mn}$ | 122 | 45,491 | 2007-10-29 | 2018-12-24 | / | 4.59 | CNEMC |
| | | | DIP | 3 | 9 | 1981-08-06 | 1983-11-27 | / | 0.00 | GRQA |
| | | | DO | 135 | 45,932 | 1980-01-07 | 2018-12-24 | / | 3.99/3.59 | CNEMC(45,459)/GRQA(473) |
| | | | DOC | 5 | 16 | 1981-07-22 | 2008-05-21 | / | 0.00 | GRQA |
| | | | DOSAT | 24 | 31 | 1986-01-14 | 1999-02-11 | / | 3.23 | GRQA |
| | | | NH4N | 123 | 45,567 | 1983-02-24 | 2018-12-24 | / | 12.28/0.00 | CNEMC(45,562)/GRQA(5) |
| | | | NO2N | 13 | 334 | 1981-08-06 | 1997-11-10 | / | 7.19 | GRQA |
| | | | NO3N | 119 | 388 | 1981-07-22 | 2009-09-05 | / | 6.96 | GRQA |
| | | | pH | 251 | 46,181 | 1980-01-21 | 2018-12-24 | / | 0.50/0.99 | CNEMC(45,571)/GRQA(610) |
| | | | TDP | 3 | 16 | 1994-04-12 | 1996-10-21 | / | 0.00 | GRQA |

| | | | | | | | | | |
|---|---|---|---|---|---|---|---|---|---|
| | | | TEMP | 92 | 520 | 1980-02-06 | 2009-04-05 | / | 0.00 | GRQA |
| | | | TOC | 1 | 1 | 1994-08-30 | 1994-08-30 | / | 0.00 | GRQA |
| | | | TP | 10 | 196 | 1985-01-07 | 1996-10-17 | / | 15.31 | GRQA |
| | | | TSSs | 12 | 329 | 1980-01-08 | 1997-09-22 | / | 9.73 | GRQA |
| Lake | 22 | 4 | $COD_{Mn}$ | 22 | 6657 | 2007/10/29 | 2018/12/24 | / | 10.64 | CNEMC |
| | | | DO | 22 | 6656 | 2007/10/29 | 2018/12/24 | / | 2.48 | CNEMC |
| | | | NH4N | 22 | 6667 | 2007/10/29 | 2018/12/24 | / | 6.90 | CNEMC |
| | | | pH | 22 | 6661 | 2007/10/29 | 2018/12/24 | / | 0.05 | CNEMC |
| Reservoir | 5 | 4 | $COD_{Mn}$ | 5 | 2231 | 2007/10/29 | 2018/12/24 | / | 8.70 | CNEMC |
| | | | DO | 5 | 2276 | 2007/10/29 | 2018/12/24 | / | 1.36 | CNEMC |
| | | | NH4N | 5 | 2268 | 2007/10/29 | 2018/12/24 | / | 11.02 | CNEMC |
| | | | pH | 5 | 2252 | 2007/10/29 | 2018/12/24 | / | 0.27 | CNEMC |

Availability (Figure 4a) and continuity (Figure 4b) plots were used to examine the temporal fragmentation of the time series. Some dominated indicators (i.e., $COD_{Mn}$, DO, NH4N, pH) were selected to present in Figure 4. Our analysis revealed that observations from inland exhibited significantly higher availability and continuity than ocean. Specifically, for weekly water quality data, data availability for all indicators ranged from 40% to 80% (Figure 4a), indicating good data availability. In contrast, observations from the ocean showed moderate availability while exhibited low data continuity for most observations.

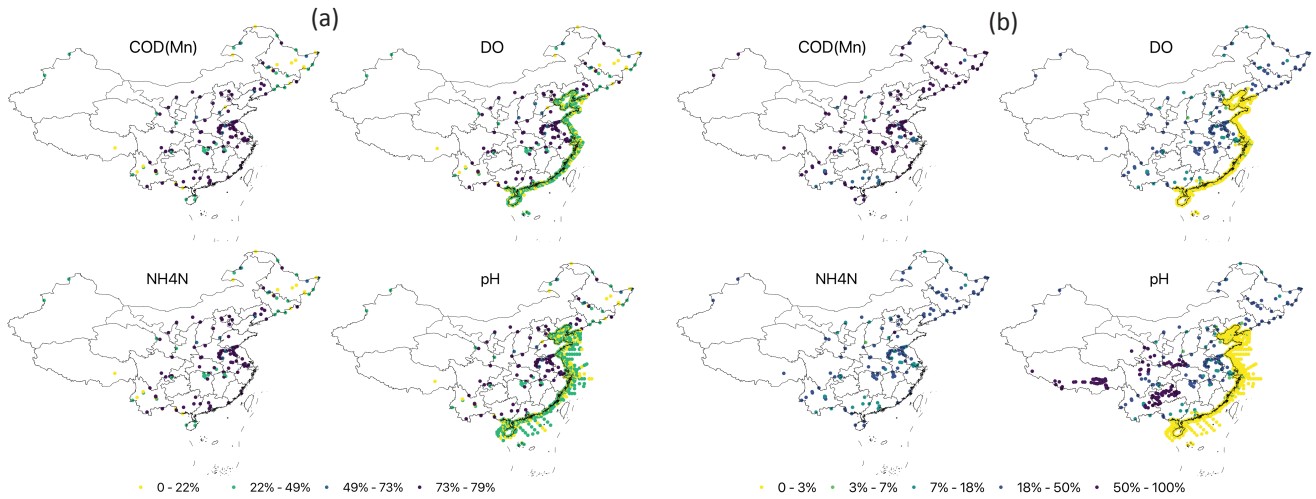

**Figure 4. Overall availability (a) and continuity (b) for KMnO₄ chemical oxygen demand ($COD_{Mn}$), dissolved oxygen (DO), ammonia nitrogen (NH4N), and pH.**

The presentation of outlier proportions was documented in Table 3. Among all indicator types, TP and NH4N exhibited a higher proportion of outliers (Table 3). After the removal of outliers detected through the IQR test, boxplots were constructed for each indicator, illustrating a prominent positive skew in their distributions (Figure 5). However, in the case of the TOC

indicator, the generation of a boxplot was not informative due to the presence of only a single data point (Table 3), and as such, it was omitted from presentation in this context. This skewness behavior was consistent with the characteristics observed in the GRQA dataset. Conversely, indicators of DO and pH demonstrated a significant normal distribution across all three data sources.

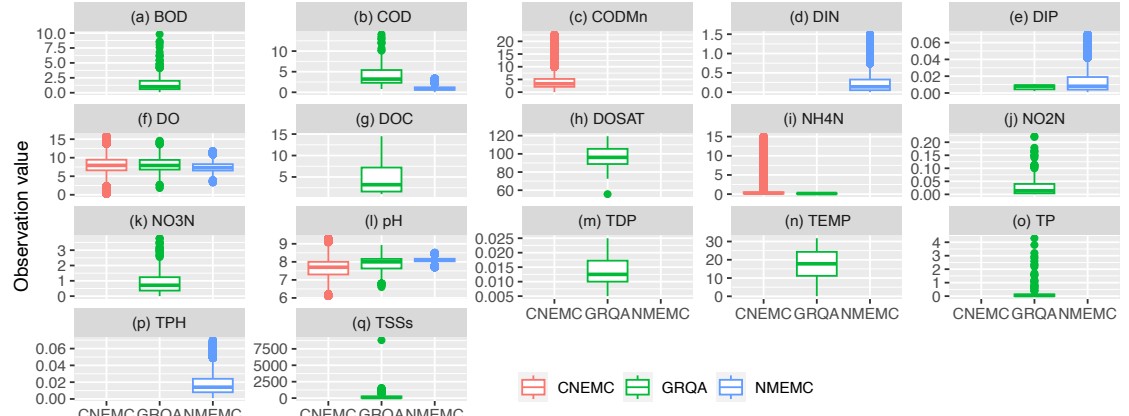

**Figure 5. Boxplots for all indicators with (a) biochemical oxygen demand (BOD), (b) chemical oxygen demand  (COD), (c) KMnO₄ chemical oxygen demand (COD$_{Mn}$), (d) dissolved inorganic nitrogen (DIN), (e) dissolved inorganic phosphorus (DIP), (f) dissolved oxygen (DO), (g) dissolved organic carbon (DOC), (h) dissolved oxygen saturation (DOSAT), (i) ammonia nitrogen (NH4N), (j) nitrite nitrogen (NO2N), (k) nitrate nitrogen (NO3N), (l) potential of hydrogen (pH), (m) total dissolved phosphorus (TDP), (n) temperature (TEMP), (o) total phosphorus (TP), (p) total petroleum hydrocarbons (TPH), and (q) total suspended solids (TSSs)). Outliers determined by the interquartile range (IQR) has been removed. The unit of indicators except TEMP (∘C), pH (%), and DOSAT (%) were mg L⁻¹.**

## 4 Applications

Given the amount of metadata information included in our inventory and the observations, this database will be particularly useful and important for researchers and decision-makers in the fields of hydrology, environmental research, water resources management, ecological studies, climate change, policy development, public health, and oceanography. For example, the indicator of NH4N can be used by hydrologists to develop predictive models, calibrate nitrogen models, and generate projections within China. The inland and coastal/oceanic water quality data can be connected to display the dynamic of water quality from land to ocean, thereby routing the import, transport, and export of pollutants. Researchers can use this data to analyze long-term trends and variations in surface water quality, which can be vital for understanding the impact of various factors such as climate change, pollution, and land use on aquatic ecosystems. Water resource managers can utilize this repository to assess the quality of water in different regions, helping to make informed decisions about water allocation, treatment, and conservation strategies. Policymakers can rely on this repository to support evidence-based policy development

related to water quality standards and regulations. Health officials can use this data to monitor the safety of water sources and assess potential health risks associated with waterborne contaminants. The high intensity of coastal/oceanic water quality data can be used to indicate coastal/oceanic water environment for food web (i.e., living conditions of plankton). For instance, phytoplankton and zooplankton communities are sensitive to the changes in water quality, and respond to low DO levels, high 350 nutrient levels (i.e., DIN), and toxic contaminants (i.e., TPH). Therefore, such spatial continuous coastal/oceanic water quality dataset is helpful for characterizing the patterns of spatial-temporal distributions of plankton, assessing the status and trends of biodiversity, and predicting the population succession in the changing ocean world.

Certain studies have previously utilized specific segments of the original dataset. For instance, researchers have employed the weekly water quality data to examine the characteristics, trends, and seasonality of water quality in the Yangtze River (Di et 355 al., 2019; Duan et al., 2018). It should be noted, however, that the complete dataset presented in this study has not been employed in any research thus far, which may limit the reliability of the dataset. In future, we plan to employ this dataset in upcoming research projects, where we will rigorously test its reliability.

## 5 Data availability

All data records can be found via the figshare repository at https://doi.org/10.6084/m9.figshare.22584742.v1 (Lin et al., 360 2023a).

## 6 Conclusions

This water quality dataset was developed with the express purpose of addressing the substantial demand for Chinese water quality data, facilitating the enhancement of national water data sharing initiatives, and fostering advancements in global water-related research and applications. It provided a clean, editable, and sharable national water quality dataset within China, 365 compiling three publicly available (sub)datasets from GRQA, CNEMC, and NMEMC. The current dataset included water quality data at 2384 sites for daily at 244 sites, weekly at 149 sites, and monthly at 1991 sites in the period of 1980-2022, with over 330,000 observations for 18 indicators across both inland and coastal/oceanic domains. The predominant share of observations, comprising approximately 98.9%, originates from the CNEMC and NMEMC, significantly expanding the global water quality dataset with a notable emphasis on the Asian region.

This database will be particularly useful and important for researchers and decision-makers in the fields of hydrology, environmental management, and oceanography for advancing the assessment, modeling, and projection of water quality, ocean biomass, and biodiversity in China. Considering the extensive coverage of oceanic monitoring sites within this dataset, it has made a substantial contribution to the dissemination of coastal/oceanic water quality data, offering a comprehensive depiction of the aquatic environment, and facilitating researchers in conducting in-depth investigations into ocean ecosystem. Due to its

comprehensive temporal coverage of riverine water quality data, this dataset presented a valuable adjunct for research requiring large sample datasets and continuous information, especially for watershed modeling, such as water pollutants modeling and projection.

This water quality dataset will be regularly updated to incorporate any new publicly released government data in China, ensuring prompt availability to the community for their immediate use. Considering the existing absence of biological parameters within the global water quality dataset, we have the intention to proactively incorporate relevant biological parameters in the event of new government data releases. This dataset also introduces the metadata framework for forthcoming national datasets, a comprehensive collection of water-related data throughout China that aims at providing free, clean, non-sensitive, coherent, and reliable water data within China for global researchers to support the national water resources management and further promote Asian water data sharing in the future.

## Competing interests

The contact author has declared that none of the authors has any competing interests.

## Acknowledgements

This work was supported by the Program for Guangdong Introducing Innovative and Entrepreneurial Teams (grant number 2019ZT08L213) and the National Natural Science Foundation of China (grant number U20A20117, 52200213, 52239005).

## Author contributions

Yanpeng Cai and Zhifeng Yang were involved in planning and supervised the work. Jingyu Lin, Peng Wang, and Jinzhu Wang designed the code. Jingyu Lin carried out the data processing with contributions from Peng Wang and Jinzhu Wang. Jingyu Lin mapped the monitoring sites and developed the outlier detection strategy. Youping Zhou helped improve the grammar and flow of the manuscript. Jingyu Lin prepared the manuscript and Jinzhu Wang, Youping Zhou, Xudong Zhou, Pan Yang, and Hao Zhang provided critical feedback and helped shape the research, analysis, and manuscript.

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
