# Peer review of "An extensive spatiotemporal water quality dataset covering four decades (1980-2022) in China"

_Earth System Science Data, 2023_

## Author Comment (AC1)

Reply letter to all reviewers of the preprint submitted to Earth System Science Data (ESSD) entitled: "Water quality dataset in China".

The authors' answers highlighted in blue are given below the reviewer's comments. The updated version of the paper tracked with changes is available from the attachment.

Community comments
1. The paper aims to improve availability of water quality data in China by adding weekly and 3-montly averages to a global water quality dataset GRQA (Virro et al., 2021). The weekly and 3-monthly averages were extracted from pdf-s that raises additional data quality issues. The authors needed to geocode the points semi-automatically and validate against stream and watershed datasets. Water quality data is quite scarce and therefore any attempt to improve the availability of the data is highly welcome.

Reply: Thanks for your comments.

2. The paper is in general well written, clearly structured and illustrated with tables and figures sufficiently.

Reply: Thanks for your comments.

3. My main concern is that the main part of the dataset are the weekly and 3-montly water quality indicators that do not have any information about how many samples are in the averages, nor do they have any basic statistics (range, variance etc) about the original data based on which the averages have been obtained. Without having this information, the value and use of the data is severely limited. Also, how adequate is the average for water quality data? Water quality usually does not exhibit normal distribution and therefore average might be quite biased. This should be addressed in the paper.

Reply: This is a good question. It should be noted that these data were monitored, gathered, analysed, and released by the national automatic monitoring station government. What data would be published to the public greatly depends on the willingness of authority. By now, the Chinese government only make the weekly and monthly water quality data available to the public.

Follow your suggestion, we added some information of weekly water quality data in lines 145-152 "This weekly water quality data was collected and constructed by following the standards from the Environmental Quality standards for surface water (GB3838-2002). Water samples were automatically collected at six intervals throughout the day, with a sampling frequency of one sample every four hours (00:00-04:00, 04:00-08:00, 08:00-12:00, 12:00-16:00, 16:00-20:00, 20:00-24:00). The weekly water quality dataset was derived through the computation of daily averages encompassing Monday through Sunday. This process yielded a single numerical value that served as a representative of a set of valid data samples. Specifically, a minimum of four data samples were aggregated to calculate the daily average, and five daily average data points were used to compute the weekly average."

We amended the information of ocean water quality data in lines 162-167 "Guidelines in the Specification for Offshore Environmental Monitoring (HJ 442-2008) directed the methodologies, criteria, and quality assurance measures for monthly sampling of ocean water quality. Employing Niskin and Go-Flo water samplers, samples were collected multiple times annually, typically

during the months of April through December, as illustrated in Figure 1. The acquisition of this dataset entailed the collection of various quality control samples, including matrix spikes, blanks, parallels, and quality control check samples, which underwent meticulous collection and subsequent intra-laboratory comparison."

Based on what we have, the ability of our team is to tidy up and standardize the data so that the academic community can make full use of them. Even though some of this dataset is limited to the missing of original data, this dataset still can be used but not limited in the hydrology, oceanography, ecology, environment, geography, biology.

We understand your concerns of the quality of average weekly data. In Section 2.2.3, several measures were undertaken to validate the (sub)dataset, including quality control procedures and cross-validation with other datasets. In Section 3.3, an evaluation of availability and continuity for water quality data was conducted to elucidate the quality of the data. These rigorous steps collectively contribute to enhancing the reliability of the (sub)dataset. We also mentioned the previous application of this (sub)dataset in Application Section "Certain studies have previously utilized specific segments of the original dataset. For instance, researchers have employed the weekly water quality data to examine the characteristics, trends, and seasonality of water quality in the Yangtze River (Di et al., 2019; Duan et al., 2018)."

4. Moreover, I believe that the paper is more appropriate to publish in a local/regional journals or data repositories rather in the Earth Systems Science Data because it only covers data for China.

Reply: Our paper is dedicated to the reorganization and standardization of water quality data in China, addressing the substantial demand for comprehensive Chinese water data. This endeavour holds significant implications for the hydrology, environmental management, and oceanography communities. Given the current scarcity of high-quality water quality data in China, our dataset is poised to attract keen interest from researchers and managers alike. Its subsequent utilization is expected to make substantial contributions to the field of Earth system sciences.

This initiative aligns seamlessly with the objectives of the Earth Systems Science Data journal, which has a history of publishing local data repositories, including water quality monitoring data from various regions, such as the United Kingdom (Bowes et al., 2018), Germany (Ebeling et al., 2022), and Arctic watersheds (Shogren et al., 2022). Therefore, the geographic scope of our dataset, which is limited to China, should not impede its potential academic contributions to the global scientific community.

5. The compilation of the dataset is partly not sufficiently described, and it is not possible to fully understand based on which criteria the authors decided to include/exclude some measurements or recode. Please see my additional comments on this in the attached file.

Additional comments:

- 1. "and President Xi's version of Chinese Dream" I think that this is not relevant for the international community and therefore I recommend to remove it.

Reply: Removed accordingly.

- 2. "2.2.2 Metadata information processing" In this section, you describe adding coordinates to the water quality data and therefore define coordinates as metadata. I disagree with coordinates being metadata as they are part of your dataset not data about data. Please rename this section e.g. "Coordinates of the monitoring sites" and correct the wording in the section.

Reply: Thanks. We amended the section and corrected the words for the whole section.

- 3. "duplicated and irrelevant rows were" how were duplicates identified? What were irrelevant rows?

Reply: We described it with "In addition, duplicated rows were identified and removed by using distinct function in R based on the unique site, indicators, monitoring week/date, and values from the (sub)datasets that included 1776 site pairs from the weekly water quality dataset due to the file inconsistencies mentioned in 2.2.1." Irrelevant rows refer to descriptive rows which was detailed in lines 257-259 "They were validated with the descriptive text on the cover of each report that was deleted later from the weekly water quality dataset."

- 4. "messed" do you mean merged ?

Reply: yes, we revised it accordingly.

- 5. "*No detected*" what does "no detected" mean? Do you mean "not dtected" and under taht you mean that the values were below the detection limit? If so, what was the detection limit?

Reply: We have clarified "Values that falling below known detection limits were labelled as "< DL" from the monthly water quality datasets. COD, DO, DIN, DIP, and TPH detection limits were 0.15 mg/L, 0.32 mg/L, 0.001 mg/L, 0.001 mg/L, and 0.001 mg/L, respectively."

- 6. "河南信阳徐桥"what does that mean?

Reply: It is the name of a station in Chinese from the original files. We further clarified the meaning of"河南信阳徐桥"with its Hanyu Pinyin in lines 218-220 "As the geographic coordinates for the station labeled "Xuqiao" were unidentifiable from the provided information within the original files, the data associated with this station were excluded from the dataset."

- 7. "We provided water quality dataset including NA value and excluding NA value for different data users." needs more explanations.

Reply: We have removed this sentence to avoid misunderstanding. We provided full datasets without NA/missing value.

6. The data must be properly deposited in an open data repository with a DOI and relevant metadata. Currently, the DOI indicated in the paper is not working.

Reply: Thanks. We have activated the DOI so that you can find it works now.

General Comment

Lin et al. derived a new dataset of surface water quality in China from three sources. Due to the limited water quality data of China in current global dataset, this dataset presented in this study represents a significant contribution to the water quality community. However, I found the current version of manuscript reads more like a technical report that documents how the dataset was derived. The authors should implement more analysis with the new dataset to demonstrate its reliability and usability. I am not asking the authors to implement novel analysis or come up with new insights on water quality based on the dataset. But I think it will be very helpful for the authors to implement more common analysis (e.g., seasonality, trending, etc.). Based on this reason, I would like to recommend a major revision before publication. Please also see additional comments in the following.
Reply: Thanks for your comment.

To address the concerns you mentioned, we made a throughout revision for our manuscript including elaborating on the data cleaning process (See Section 2.2.3) to ensure data consistency from different data sources, implementing analysis to demonstrate the spatial and temporal distribution, variation, availability, and continuity of monitoring sites, observations, and indicators (See Section 3.2 and Section 3.3).

Major Comment

Should clarify the number of sites daily, weekly, and monthly observations accordingly. The authors mentioned the observation is available for the period of 1980-2022. But I believe the temporal coverage can be very different among the sites, thus another useful metric is length of data.
Reply: Thanks for the comments.

To address your concerns, we first clarified the number of sites for daily, weekly, and monthly in the abstract "It spanned 18 distinct indicators, meticulously gathered at 2384 monitoring sites, which were further categorized as daily (244 sites), weekly (149 sites), and monthly (1,991 sites), ranging from inland locations to coastal and oceanic areas."

Subsequently, we appended two new sections (i.e., Section 3.2 Spatial-temporal distribution of monitoring sites and Section 3.3 Characteristics of time series) to illustrate the spatial-temporal coverage and fragment of the dataset in lines 297-340 "...Notably, GRQA predominantly contributes observations from monitoring sites prior to 2006, with an average of 133 observations obtained from approximately 13 sites per year, as illustrated in Figure 3a and Figure 3b. In contrast, CNEMC provides data from monitoring sites between 2007 and 2018, averaging around 126 sites per year, while NMEMC covers the period from 2017 to 2022 with an average of approximately 1249 sites per year. Despite CNEMC providing fewer monitoring sites, it consists of a comparable number of observations with an average of approximately 18,145 observations per year compared to NMEMC with an average of 19,369 observations. Comparatively, CNEMC and NMEMC datasets offer a greater number of records in comparison to GRQA. Temporal overlaps between various sources were identified on two occasions. The first instance transpired during

the years 2007 to 2009, involving data from the GRQA and the CNEMC. The second temporal overlap was documented between CNEMC and NMEMC for the years 2017 to 2018. Overall, the number of monitoring sites with records exhibited a slight increase before 2016, followed by a significant surge after 2016."

[Figure]

Figure 3. Distribution of monitoring sites (a) and observations (b) from different sources over time.

Availability (Figure 4a) and continuity (Figure 4b) plots were used to examine the temporal fragmentation of the time series. Some dominated indicators (i.e., COD$_{Mn}$, DO, NH4N, pH) were selected to present in Figure 4. Our analysis revealed that observations from inland rivers/lakes/reservoirs exhibited significantly higher availability and continuity than ocean. Specifically, for weekly water quality data, data availability for all indicators ranged from 40% to 80% (Figure 4a), indicating good data availability. In contrast, observations from the ocean showed moderate availability while exhibited low data continuity for most observations.

[Figure]

Figure 4. Overall availability (a) and continuity (b) for KMnO₄ chemical oxygen demand (COD$_{Mn}$), dissolved oxygen (DO), ammonia nitrogen (NH4N), and pH.

In addition, we counted the length of data for each time series, which was provided at *Supplementary Information Metadata and Statistics.*

Minor Comments

Line 34: Need to introduce SDG before using the acronym.
Reply: We have introduced SDGs when first mentioned in lines 38-41 "Water, constituting the foundational pillar of sustainable development (UNESCO, 2019), bears a profound interconnection with numerous targets within the Sustainable Development Goals (SDGs), notably SDG 6 (Sadoff et al., 2020), which endeavors to ensure the universal availability and sustainable management of water and sanitation".

Line 55-57: This statement is confusing. What do you mean by "different metadata information"?

Reply: We have made a major revision for the Introduction section. This statement and the attached paragraph were replaced with a new one in lines 97-103 "…Additionally, the water data available from open data centres are stored in a user-unfriendly format that require significant additional efforts to make them credible, editable, and reusable. For example, monthly water quality data spanning from 2006 to 2022 are presented as reports with figures derived from statistical analysis, instead of providing more reliable monitoring data. Although some studies have employed national-scale water quality data for assessment and modelling covering whole China (Ma et al., 2020a; Ma et al., 2020b; Huang et al., 2021; Zhang et al., 2022), these datasets are not publicly available due to licensing restrictions and/or government-sanctions (Lin et al., 2023). To date, there is no clean and publicly accessible national water quality dataset covering whole China."

Line 96: I suggest:" Data presented in this paper…".

Reply: Since the descriptions of CDWA was removed, the sentence the reviewer mentioned was dropped off at the same time. Now the whole paragraph reads "…Our water quality dataset is thus initiated to meet the huge demand for Chinese water quality data, to boost national water data sharing, and to advance global water-related research and applications. It intends to collect non-sensitive and publicly available water quality data, to apply consistency to the formatting and curation, and to establish a standardized set of metadata for different water quality aspects."

Line 90-97: In my understanding, CWDA is already a public data archive and the authors added new water quality data to this archive. If so, please focus on describing more for the water quality data that is presented in this study.

Reply: To avoid misunderstanding, the descriptions of CDWA was removed. We directed our focus towards water quality data in lines 104-116 "Therefore, there is a pressing need to reorganize, curate, and manage the continuous, long-time series, standardized, well-organized, and consistent water quality datasets from inland to coastal/oceanic areas within China. These datasets stand as invaluable resources to support researchers and decision-makers. They enable an in-depth examination of water quality status, encompassing the entire spectrum from riverine environments to the vast expanse of the oceans. Furthermore, they provide the means to model various dimensions of water quality indicators and forecast the ramifications of emergent water pollution phenomena (i.e., coastal eutrophication and oceanic harmful algal blooms due to additional nitrogen input from land and releases of radionuclides from inland redundant nuclear power plant accidents). It is also valuable to the effective management of water resources to support the United Nation Water Action Decade (2018-2028) and Ocean Decade (2021-2030; Folke et al., 2021). Our water quality dataset is thus initiated to meet the huge demand for Chinese water quality data, to boost national water data sharing, and to advance global water-related research and applications. It intends to collect non-sensitive and publicly available water quality data, to apply consistency to the formatting and curation, and to establish a standardized set of metadata for different water quality aspects."

Line 182: What does "messed into" mean? Mixed?

Reply: We clarified it as "Some observations for different indicators were merged into a single column when converting the PDF file to editable files for weekly water quality data."

Line 185: Should clarify the meaning of "未检出" and "河南信阳徐桥". And is the later the only station removed?

Reply: We have clarified the meaning of "未检出" in lines 264-266 "Values that falling below known detection limits were denoted as "< DL" within the monthly water quality datasets. COD, DO, DIN, DIP, and TPH detection limits were 0.15 mg/L, 0.32 mg/L, 0.001 mg/L, 0.001 mg/L, and 0.001 mg/L, respectively."

We further clarified the meaning of "河南信阳徐桥" with its Hanyu Pinyin in lines 218-220 "As the geographic coordinates for the station labeled "Xuqiao" were unidentifiable from the provided information within the original files, the data associated with this station were excluded from the dataset."

Line 188: Do you mean the dataset is provided with two versions?

Reply: We removed this sentence to avoid misunderstanding. Treatment of NA data was mentioned in lines 264 "Missing (e.g., noted as '-') and empty data were replaced with NA, and were omitted from the dataset."

Line 216: The statement about the outliers is ambiguous. I don't get if the authors were trying to argue the data is less impact by the outliers or not. In addition, more explanations and quantification of the outliers' number will be very helpful.

Reply: Thanks for the suggestions.

We first explained the method for detecting the outliers in lines 280-283 "Outliers were detected by using the interquartile range (IQR) method. IQR is the range between the first (Q1) and third (Q3) quartile. Data points that fell below Q1-1.5×IQR and above Q3+1.5×IQR were considered outliers. Since it was difficult to determine whether an outlier is an error caused by faulty equipment or data entry errors or not, no observations were omitted from the original datasets."

Then, we calculated the proportion of outliers for each time series that was documented in Table 3 and explained in lines 350-356 "The presentation of outlier proportions was documented in Table 3. Among all indicator types, NH4N exhibited a higher proportion of outliers (Table 3), … However, in the case of the TOC indicator, the generation of a boxplot was not informative due to the presence of only a single data point (Table 3), and as such, it was omitted from presentation in this context…"

**Table 1. Stats for different types of the monitoring sites and indicators.**

| Location Type | Sites in total | Indicators' number | Indicators' name | Sites | Observations | Start date | End date | Below limits(n) | Outliers (%) | Sources(n) |
|---|---|---|---|---|---|---|---|---|---|---|
| Coast/Ocean | 1991 | 6 | COD | 1991 | 19,367 | 2017-05 | 2022-08 | 94 | 4.88 | NMEMC |
| | | | DIN | 1991 | 19,369 | 2017-05 | 2022-08 | / | 8.99 | NMEMC |
| | | | DIP | 1991 | 19,369 | 2017-05 | 2022-08 | 939 | 6.76 | NMEMC |
| | | | DO | 1991 | 18,143 | 2017-05 | 2022-08 | / | 2.78 | NMEMC |
| | | | pH | 1991 | 19,338 | 2017-05 | 2022-08 | / | 3.69 | NMEMC |
| | | | TPH | 1991 | 19,368 | 2017-05 | 2022-08 | 2453 | 2.88 | NMEMC |
| River | 366 | 15 | BOD | 10 | 432 | 1980-01-07 | 1997-11-27 | / | 6.71 | GRQA |
| | | | COD | 10 | 235 | 1988-01-03 | 1997-11-27 | / | 6.81 | GRQA |
| | | | COD$_{Mn}$ | 122 | 45,491 | 2007-10-29 | 2018-12-24 | / | 4.59 | CNEMC |
| | | | DIP | 3 | 9 | 1981-08-06 | 1983-11-27 | / | 0.00 | GRQA |
| | | | DO | 135 | 45,932 | 1980-01-07 | 2018-12-24 | / | 3.99/3.59 | CNEMC(45,459)/GRQA(473) |

| | | | Indicator | | | | | | | |
|---|---|---|---|---|---|---|---|---|---|---|
| | | | DOC | 5 | 16 | 1981-07-22 | 2008-05-21 | / | 0.00 | GRQA |
| | | | DOSAT | 24 | 31 | 1986-01-14 | 1999-02-11 | / | 3.23 | GRQA |
| | | | NH4N | 123 | 45,567 | 1983-02-24 | 2018-12-24 | / | 12.28/0.00 | CNEMC(45,562)/GRQA(5) |
| | | | NO2N | 13 | 334 | 1981-08-06 | 1997-11-10 | / | 7.19 | GRQA |
| | | | NO3N | 119 | 388 | 1981-07-22 | 2009-09-05 | / | 6.96 | GRQA |
| | | | pH | 251 | 46,181 | 1980-01-21 | 2018-12-24 | / | 0.50/0.99 | CNEMC(45,571)/GRQA(610) |
| | | | TDP | 3 | 16 | 1994-04-12 | 1996-10-21 | / | 0.00 | GRQA |
| | | | TEMP | 92 | 520 | 1980-02-06 | 2009-04-05 | / | 0.00 | GRQA |
| | | | TOC | 1 | 1 | 1994-08-30 | 1994-08-30 | / | 0.00 | GRQA |
| | | | TP | 10 | 196 | 1985-01-07 | 1996-10-17 | / | 15.31 | GRQA |
| | | | TSSs | 12 | 329 | 1980-01-08 | 1997-09-22 | / | 9.73 | GRQA |
| Lake | 22 | 4 | CODMn | 22 | 6657 | 2007/10/29 | 2018/12/24 | / | 10.64 | CNEMC |
| | | | DO | 22 | 6656 | 2007/10/29 | 2018/12/24 | / | 2.48 | CNEMC |
| | | | NH4N | 22 | 6667 | 2007/10/29 | 2018/12/24 | / | 6.90 | CNEMC |
| | | | pH | 22 | 6661 | 2007/10/29 | 2018/12/24 | / | 0.05 | CNEMC |
| Reservoir | 5 | 4 | CODMn | 5 | 2231 | 2007/10/29 | 2018/12/24 | / | 8.70 | CNEMC |
| | | | DO | 5 | 2276 | 2007/10/29 | 2018/12/24 | / | 1.36 | CNEMC |
| | | | NH4N | 5 | 2268 | 2007/10/29 | 2018/12/24 | / | 11.02 | CNEMC |
| | | | pH | 5 | 2252 | 2007/10/29 | 2018/12/24 | / | 0.27 | CNEMC |

Finally, we analysed the distribution of observations value in lines 350-356 after the removal of outliers and making boxplots for each indicator "After the removal of outliers detected through the IQR test, boxplots were constructed for each indicator, illustrating a prominent positive skew in their distributions (Figure 5). This skewness behavior was consistent with the characteristics observed in the GRQA dataset. Conversely, indicators of DO and pH demonstrated a significant normal distribution across all three data sources."

[Figure]

**Figure 5. Boxplots for all indicators with (a) biochemical oxygen demand (BOD), (b) chemical oxygen demand (COD), (c) KMnO$_4$ chemical oxygen demand (COD$_{Mn}$), (d) dissolved inorganic nitrogen (DIN), (e) dissolved inorganic phosphorus (DIP), (f) dissolved oxygen (DO), (g) dissolved organic carbon (DOC), (h) dissolved oxygen saturation (DOSAT), (i) ammonia nitrogen (NH4N), (j) nitrite nitrogen (NO2N), (k) nitrate nitrogen (NO3N), (l) potential of hydrogen (pH), (m) total dissolved phosphorus (TDP), (n) temperature (TEMP), (o) total phosphorus (TP), (p) total petroleum hydrocarbons (TPH), and (q) total suspended solids (TSSs)). Outliers determined by the interquartile range (IQR) has been removed. The unit of indicators except TEMP ($\circ$C), pH (%), and DOSAT (%) were mg L$^{-1}$.**

Figure 4: I think it is better to use different color to represent the sites from different sources.

Reply: We amended it accordingly. Now it looks

[Figure]

**Figure 2. Spatial distribution of water quality monitoring sites from different sources with drainages in China.**

This paper reconstructed the historical water quality data in inland, coastal and ocean areas of China. This dataset would be useful for further water quality related research in China. However, this paper does not appily the dataset to any researches and the reliability of the dataset does not be proved. Overall, this manuscript is clearly organized, but I think this manuscript should be reconsidered after major revision.
Reply:

We appreciate the reviewer's observation regarding the absence of research applications in our paper. The decision to refrain from applying the dataset in this study was intentional and based on the scope and objectives of our work. Our primary aim in this paper was to present and describe the dataset comprehensively, including its sources, data collection methods, and harmonization processes. We intended to provide a valuable resource for the scientific community, researchers, and decision-makers interested in utilizing this dataset for various research applications.

We have acknowledged in the paper's Application Section that the dataset has not been utilized for research purposes "Certain studies have previously utilized specific segments of the original dataset. For instance, researchers have employed the weekly water quality data to examine the characteristics, trends, and seasonality of water quality in the Yangtze River (Di et al., 2019; Duan et al., 2018). It should be noted, however, that the complete dataset presented in this study has not been employed in any research thus far, which may limit the reliability of the dataset. In future, we plan to employ this dataset in upcoming research projects, where we will rigorously test its reliability."

Furthermore, in Section 2.2.3, several measures were undertaken to validate the dataset, encompassing quality control procedures and cross-validation with other datasets. In Section 3.3, an evaluation of availability and continuity for water quality data was conducted to elucidate the quality of the data. These rigorous steps collectively contribute to enhancing the reliability of our dataset.

We hope this clarifies our approach and addresses the reviewer's concerns regarding the non-application of the dataset in this paper.

Specific comments

Line 39-40: "Amongst the water quality data" what "is a key aspect used...", or you want to say "water quality data is a key aspect..."
Reply: This sentence was clarified as "Within the context of the Source-to-Sea (S2S) aquatic continuum, water quality data emerges as a pivotal factor in discerning pollution levels (Regnier et al., 2022)."

Table 2: "spatial resolution" to "Spatial resolution"
Reply: Revised accordingly.

The study introduces a water quality dataset for China by reorganizing and consolidating data from various sources, including the Global River Water Quality Archive (Virro et al. 2021), China National Environmental Monitoring Centre, and National Marine Environmental Monitoring Center. The dataset holds significant potential interest for the community; however, the manuscript's overall quality is low. I recommend that the authors undertake a comprehensive revision of the manuscript before proceeding to its resubmission.

The Introduction section would benefit from a thorough rewrite, while the Data & Methodology section should be augmented with additional details. Moreover, the Results section should encompass independent validation and dataset intercomparison. It is recommended to include a foundational analysis of basic consistency or continuity, thus substantiating the reliability of the processing undertaken. Lastly, meticulous attention to English grammar should be given during the manuscript's revision process.

Specifically,

1. Introduction Section: The current presentation of the introduction begins with a discussion of water data, yet it lacks a central focus on water quality. Notably absent are clear definitions of water quality indicators with their potential significance. To enhance this section, I propose a restructuring along the following lines:

a. Establish a fundamental academic context surrounding water quality, incorporating key indicators that are widely recognized.

b. Emphasize the critical importance of maintaining high water quality standards across various domains.

c. Address the existing landscape of water quality datasets and their application examples, highlighting the shortcomings.

d. Convey the distinctive innovations and contributions that this study brings to the field.

This will lend greater clarity and engagement to the introduction, better aligning it with the study's objectives and significance.

Reply: We highly appreciated your very specific comments on restructuring the Introduction Section. Follow your proposal, we restructured and rewrote the whole section in lines 49-116"

[revised manuscript text omitted]
 Section: Given that the raw data was collected rather than generated by this study, please provide additional details and context for the original datasets, such as sensors, quality maintenance methods, etc.

Reply: We added this information for the weekly water quality data in lines 144-151 "This weekly water quality data was collected and constructed by following the standards from the Environmental Quality standards for surface water (GB3838-2002). Water samples were automatically collected at six intervals throughout the day, with a sampling frequency of one sample every four hours (00:00-04:00, 04:00-08:00, 08:00-12:00, 12:00-16:00, 16:00-20:00, 20:00-24:00). The weekly water quality dataset was derived through the computation of daily averages encompassing Monday through Sunday. This process yielded a single numerical value that served as a representative of a set of valid data samples. Specifically, a minimum of four data samples were aggregated to calculate the daily average, and five daily average data points were used to compute the weekly average."

Additional details for the monthly water quality data were given in lines 161-166 "Guidelines in the Specification for Offshore Environmental Monitoring (HJ 442-2008) directed the methodologies, criteria, and quality assurance measures for monthly sampling of ocean water quality. Employing Niskin and Go-Flo water samplers, samples were collected multiple times annually, typically during the months of April through December, as illustrated in Figure 1. The acquisition of this dataset entailed the collection of various quality control samples, including matrix spikes, blanks, parallels, and quality control check samples, which underwent meticulous collection and subsequent intra-laboratory comparison."

3. Methodology: It's imperative to elaborate on the data cleaning process. Explain the methods employed to remove abnormal values and ensure data consistency from different data sources.

Reply: We introduced the data cleaning and harmonization process in a rewritten Section 2.2.3 in lines 234 – 267 "

[revised manuscript text omitted]

5. Dataset Assessment: Present comprehensive assessments of the dataset, including its spatial and temporal consistency. Address questions regarding spatiotemporal overlap between data sources and the congruence of processed outputs from different sources.
Reply: We first undertook a comprehensive standardization process across all data sources. This harmonization encompassed the transformation of downloaded time series into a uniform file format, shifting from CSV files to R time series. Additionally, we ensured consistency in indicator selection, units, data structure, identification of missing values, and language.

Considering the questions of spatiotemporal overlap between data sources, we have identified them and clarified in lines 297-298 "Following cross-validation, it was observed that there was no spatial convergence among monitoring sites from different data sources.", lines 314-316 "Temporal overlaps between various sources were identified on two occasions. The first instance transpired during the years 2007 to 2009, involving data from the GRQA and the CNEMC. The second temporal overlap was documented between CNEMC and NMEMC for the years 2017 to 2018." Given the absence of spatial overlap among monitoring sites, there is no requirement to filter observations from diverse sources.

Moreover, we assessed the availability and continuity for time series with Figure 4 for 4 dominated indicators in lines 340-345 "Availability (Figure 4a) and continuity (Figure 4b) plots were used to examine the temporal fragmentation of the time series. Some dominated indicators

(i.e., COD(Mn), DO, NH4N, pH) were selected to present in Figure 4. Our analysis revealed that observations from inland rivers/lakes/reservoirs exhibited significantly higher availability and continuity than ocean. Specifically, for weekly water quality data, data availability for all indicators ranged from 40% to 80% (Figure 4a), indicating good data availability. In contrast, observations from the ocean showed moderate availability while exhibited low data continuity for most observations."

[Figure]

**Figure 4. Overall availability (a) and continuity (b) for KMnO4 chemical oxygen demand (CODMn), dissolved oxygen (DO), ammonia nitrogen (NH4N), and pH.**

6. Language and Grammar: Carefully edit and proofread the manuscript for English grammar and language usage.

Reply: We have carefully edited and proofread the manuscript for English grammar and rephased with professional English.

Technical Issues

Line 34: 'SDG' should be clarified.
Reply: We have introduced SDGs when first mentioned in lines 38-40 "Water, constituting the foundational pillar of sustainable development (UNESCO, 2019), bears a profound interconnection with numerous targets within the Sustainable Development Goals (SDGs), notably SDG 6 (Sadoff et al., 2020)."

Line 37: "China aims at maintaining water resources while improving resources management. To achieve the United Nation's SDGs and President Xi's version of Chinese Dream, it is important to compile water data from inland to coastal/ocean areas" -> China is committed to the preservation of water resources while simultaneously advancing resource management methodologies. To effectively accomplish the United Nations' Sustainable Development Goals (SDGs) and align with China's comprehensive policy plan, it is crucial to systematically compile water-related data across both inland and coastal/oceanic domains.
Reply: Revised accordingly.

Line 39: "Amongst the water quality data is a key aspect used to identify the pollutions in the Source-to-Sea (S2S) aquatic continuum for sustaining water resources and sanitation services" -> Within the context of the Source-to-Sea (S2S) aquatic continuum, water quality data emerges as a pivotal factor in discerning pollution levels. This information plays a critical role in the preservation of water resources and the provision of sanitation services.
Reply: Revised accordingly.

Line 42: what does 'accelerated dataset' mean here?
Reply: This sentence was rephased as "The recognition of the significance of the water quality to nature, society, food, and security has accelerated the arising and availability of local, national, and global water quality datasets."

Line 45: The inclusion of Chinese water quality data within the comprehensive global dataset is notably limited, and there is a notable absence of data originating from coastal and oceanic regions.
Reply: This sentence was edited "As the largest country in East Asia, China's water quality data are notably limited in the comprehensive global dataset, with a notable absence of data from coastal and oceanic regions."

Line 54: Besides -> Moreover
Reply: Revised accordingly.

I won't continue editing the sentence but I strongly the authors utilize professional English editing to revise the manuscript.

Line 60: "if..." then what?
Reply: This whole paragraph was removed.

Line 65: this paragraph introduces several papers that were withdrawn without proving the corresponding reference or links. The writing here is more like telling stories rather than an academic paper review. The authors should pay attention to the data and review the previous datasets, applications, and drawbacks, and finally focus on stating the contributions of this work.
Reply: Thanks for your suggestion. We have removed this paragraph and reviewed the data and previous datasets in lines 77-91 "

The recognition of the significance of the water quality to nature, society, food, and security has accelerated the arising and availability of local, national, and global water quality datasets. For example, local water quality datasets include the water QUAlity, DIscharge and Catchment Attributes providing data for 1386 German catchments covering the species of nitrogen,

phosphorus, and organic carbon (Ebeling et al., 2022), a set of water chemistry measurements including carbon species, dissolved nutrients, and major ions to describe the biogeochemical conditions of permafrost-affected in Arctic watersheds (Shogren et al., 2022), catchment-wide biogeochemical monitoring platform for capturing water temperature, pH, alkalinity, suspended solid, chlorophyll concentrations, and nutrient and cation data of the Thames basin in the United Kingdom (Bowes et al., 2018). The Water Quality Portal (WQP) is comprising thousands of water quality variables encompassing physical conditions, chemical and bacteriological water analyses, chemical analyses of fish tissue, taxon abundance data, toxicity data, habitat assessment scores, and biological index scores, spanning groundwater, inland, and coastal waters, and dating back over a century (Read et al., 2017). Aggregating five large water quality datasets, the Global River Water Quality Archive (GRQA) has significantly expanded both the geographic and historical reach of existing water quality datasets by incorporating 42 parameters related to nutrient species, carbon content, sediment composition, and oxygen levels (Virro et al., 2021).

Despite significant advances in open data science for water quality research globally, Asia lags far behind other regions in this regard (Virro et al., 2021; Lin et al., 2023). As the largest country in East Asia, China's water quality data are notably limited in the comprehensive global dataset, with a notable absence of data from coastal and oceanic regions. The publicly available data consists of only 3595 daily observations in total from 244 sites, spanning from 1980 to 2009, as documented in GRQA. This is far from being adequate for water quality analysis and modelling. Additionally, the water data available from open data centres are stored in a user-unfriendly format that require significant additional efforts to make them credible, editable, and reusable. For example, monthly water quality data spanning from 2006 to 2022 are presented as reports with figures derived from statistical analysis, instead of providing more reliable monitoring data. Although some studies have employed national-scale water quality data for assessment and modelling covering whole China (Ma et al., 2020a; Ma et al., 2020b; Huang et al., 2021; Zhang et al., 2022), these datasets are not publicly available due to licensing restrictions and/or government-sanctions (Lin et al., 2023). To date, there is no clean and publicly accessible national water quality dataset covering whole China."

**Line 185: those characters are not explained in English.**
Reply: We have explained them in the Introduction part in lines 49-62"

Water quality refers to the selected physical, chemical, and biological characteristics of water that determine its suitability for a particular use (World Health Organization, 2017). There are some key properties widely recognized for measuring water quality. In terms of physical characteristics, key considerations include the color, temperature (TEMP), sediment content, turbidity, electrical conductivity, and the concentration of Total Suspended Solids (TSSs) (Oteng-Peprah et al., 2018). Chemical constituents play a significant role in the determination of water quality. These encompass parameters such as the Potential of Hydrogen (pH), acidity levels, and indicators reflecting nutrient levels, including Ammonia Nitrogen (NH4N), Nitrite Nitrogen (NO2N), and Nitrate Nitrogen (NO3N), and various forms of phosphorus such as Dissolved Inorganic Phosphorus (DIP) and Total Phosphorus (TP). Additionally, the concentration of oxygen required for microorganisms to decompose organic matter is highly considered, which includes Biochemical Oxygen Demand (BOD), Chemical Oxygen Demand (COD), and Dissolved Oxygen (DO) (Hassan Omer, 2020). Biological indicators provide insights into the presence, condition, and abundance of various living organisms within water bodies, such as bacteria, algae, and pathogens. Overall, these indicators are crucial for assessing water quality and ensuring the health of aquatic ecosystems and human populations that rely on clean water sources."

Line 202: This reference is missing from the reference list, suggest double-checking the whole manuscript to prevent it from such issues again.

Reply: As a result of substantial revisions in this section, the initial reference has been removed. We have diligently reviewed and validated both references and in-text citations.

Figure 4: all points at the coastal are clustered, suggest including regional maps to show the points clearly; and then mark the physical locations of all regional maps on a national map that can be drawn smaller than the current version. The sites from different data sources should be marked with different colors.

Reply: We have revised the map according to your suggestions.

[Figure]

**Figure 2. Spatial distribution of water quality monitoring sites from different sources with drainages in China.**

Abstract: The doi is not working, and the proposed dataset link and data reference should be provided in the abstract, please double-check the policy of ESSD.

Reply: We appended the dataset link and data reference in the abstract "This water quality dataset and supplementary metadata are available for download on figshare repository at https://figshare.com/s/4f4af7fa7b8457467ea7 (Lin et al., 2023)."

Conclusion and reference list: conclusion is too general and referred papers are limited, which makes the manuscript quality even lower.

The paper is well organized and the methods for producing this dataset is described clearly. In particular, this dataset is urgently required by researchers in environmental science, climate change, biogeochemical cycle …. I recommend to accept this manuscript after a minor revision.
Reply: Thank you for your positive feedback and valuable comments on our manuscript. We are pleased to hear that you found the paper well-organized and the methods for dataset creation clear. We also appreciate your recognition of the dataset's significance in the fields of environmental science, climate change, and biogeochemical cycles.

We will certainly address any minor revisions you suggest to further enhance the quality of the manuscript. Your insights are invaluable, and we are committed to ensuring that the paper meets the highest academic standards. We will promptly work on the recommended revisions and resubmit the manuscript accordingly.

Please see my specific comments below:

L27-28: I suggest to change the original text to "it included daily, weekly, and monthly water quality data in the period of 1980-2022, with over 330,000 observations for 17 indicators at 2384 sites from inland to coastal/ocean areas."
Reply: According to your suggestions, this sentence was clarified ", this repository comprised over 330,000 observations encompassing daily (3,588), weekly (217,751), and monthly (114,954) records of surface water quality spanning the period from 1980 to 2022. It spanned 18 distinct indicators, meticulously gathered at 2384 monitoring sites, which were further categorized as daily (244 sites), weekly (149 sites), and monthly (1,991 sites), ranging from inland locations to coastal and oceanic areas."

L29: change the 'works' to 'studies'
Reply: Revised accordingly.

L34: Give an explanation on "SDG" (full name)
Reply: We have introduced SDGs when first mentioned in lines 38-40 "Water, constituting the foundational pillar of sustainable development (UNESCO, 2019), bears a profound interconnection with numerous targets within the Sustainable Development Goals (SDGs), notably SDG 6 (Sadoff et al., 2020)."

L42: Recognition of importance of aquatic systems to ** has accelerated the arising of local and national water datasets, for example, datasets for United States *.
Reply: We have amended it as "The recognition of the significance of the water quality to nature, society, food, and security has accelerated the arising and availability of local, national, and global water quality datasets."

L53: covering China or covering whole China
Reply: This sentence was clarified "Although some studies have employed national-scale water quality data for assessment and modeling covering China (Ma et al., 2020a; Ma et al., 2020b; Huang et al., 2021; Zhang et al., 2022), these datasets are not publicly available due to licensing restrictions and/or government-sanctions (Lin et al., 2023). To date, there is no clean and publicly accessible national water quality dataset covering whole China."

L63: delete "there are"
Reply: Revised accordingly. Now it reads "Although some studies have employed national-scale water quality data for assessment and modeling covering China (Ma et al., 2020a; Ma et al., 2020b; Huang et al., 2021; Zhang et al., 2022), these datasets are not publicly available due to licensing restrictions and/or government-sanctions (Lin et al., 2023). To date, there is no clean and publicly accessible national water quality dataset covering whole China."

L64-65: these datasets are not publicly available **
Reply: Revised. Now it reads "…, these datasets are not publicly available due to licensing restrictions and/or government-sanctions (Lin et al., 2023)."

L109: spanning over period 1898-2020, or spanning from 1898 to 2020.
Reply: We have revised it with "spanning from 1898 to 2020".

L165: which converted ***, we validated **
Reply: We have amended them "We first used geocoding API methods to find the address for a given place, thereby transforming the address into a corresponding geographic entity. Afterwards, we validated each of them by overlapping with the layers of watersheds and rivers according to the official maps obtained from the National Geomatics Center of China."

L176: ** a single table and then imported into ArcGIS **
Reply: Revised accordingly.

Fig. 1a & 2a are confusing. What does the black line means? Does it denote the cumulative percentage of the missing values? The bars denote the percentage of missing values or the number of missing values? What does the right y-axis means?
Reply: These two figures were replaced with the spatial-temporal analysis of data availability and continuity after removing the missing data in lines 342-358. The length of the observation, data intensity, overall availability, longest availability, and continuity can give more details for the missing data compared to previous analysis.

"Availability (Figure 4a) and continuity (Figure 4b) plots were used to examine the temporal fragmentation of the time series. Some dominated indicators (i.e., $COD_{Mn}$, DO, NH4N, pH) were

selected to present in Figure 4. Our analysis revealed that observations from inland rivers/lakes/reservoirs exhibited significantly higher availability and continuity than ocean. Specifically, for weekly water quality data, data availability for all indicators ranged from 40% to 80% (Figure 4a), indicating good data availability. In contrast, observations from the ocean showed moderate availability while exhibited low data continuity for most observations.

[Figure]

**Figure 4. Overall availability (a) and continuity (b) for KMnO$_4$ chemical oxygen demand (COD$_{Mn}$), dissolved oxygen (DO), ammonia nitrogen (NH4N), and pH.**

We also analysed the distribution of observations value in lines 350-356 after the removal of outliers and making boxplots for each indicator "After the removal of outliers detected through the IQR test, boxplots were constructed for each indicator, illustrating a prominent positive skew in their distributions (Figure 5). This skewness behavior was consistent with the characteristics observed in the GRQA dataset. Conversely, indicators of DO and pH demonstrated a significant normal distribution across all three data sources."

[Figure]

**Figure 5. Boxplots for all indicators with (a) biochemical oxygen demand (BOD), (b) chemical oxygen demand (COD), (c) KMnO$_4$ chemical oxygen demand (COD$_{Mn}$), (d) dissolved inorganic nitrogen (DIN), (e) dissolved inorganic phosphorus (DIP), (f) dissolved oxygen (DO), (g) dissolved organic carbon (DOC), (h) dissolved oxygen saturation (DOSAT), (i) ammonia nitrogen (NH4N), (j) nitrite nitrogen (NO2N), (k) nitrate nitrogen (NO3N), (l) potential of hydrogen (pH), (m) total dissolved phosphorus (TDP), (n) temperature (TEMP), (o) total phosphorus (TP), (p) total petroleum hydrocarbons (TPH), and (q) total suspended solids (TSSs)). Outliers determined by the interquartile range (IQR) has been removed. The unit of indicators except TEMP (∘C), pH (%), and DOSAT (%) were mg L⁻¹.**

Fig. 3: Please provide a title with unit of the y-axis, and also the number e.g. a, b, c, ... for each sub-plot.

Reply: Due to the presence of multiple units for these indicators, we have provided clarification in the caption and designated each sub-plot with a series of letters (a, b, c, ...) as shown below

[Figure]

**Figure 5. Boxplots for all indicators with (a) biochemical oxygen demand (BOD), (b) chemical oxygen demand (COD), (c) KMnO$_4$ chemical oxygen demand (COD$_{Mn}$), (d) dissolved inorganic nitrogen (DIN), (e) dissolved inorganic phosphorus (DIP), (f) dissolved oxygen (DO), (g) dissolved organic carbon (DOC), (h) dissolved oxygen saturation (DOSAT), (i) ammonia nitrogen (NH4N), (j) nitrite nitrogen (NO2N), (k) nitrate nitrogen (NO3N), (l) potential of hydrogen (pH), (m) total dissolved phosphorus (TDP), (n) temperature (TEMP), (o) total phosphorus (TP), (p) total petroleum hydrocarbons (TPH), and (q) total suspended solids (TSSs)). Outliers determined by the interquartile range (IQR) has been removed. The unit of indicators except TEMP (∘C), pH (%), and DOSAT (%) were mg L$^{-1}$.**

L256-257: with 330,000 observations for 17 indicators at 2384 sites.

Reply: revised accordingly.

Editorial Comments
(1)This current dataset looks interesting, but would be strengthened by a stronger 'application' and 'validation' section.

Reply: Thanks for your interest. We have expanded our Application Section with more cases and limitations in lines 370-385

"Given the amount of metadata information included in our inventory and the observations, this database will be particularly useful and important for researchers and decision-makers in the fields of hydrology, environmental management, and oceanography. For example, the indicator of NH4N can be used by hydrologists to calibrate water quality models and generate projections within China. The inland and coastal/oceanic water quality data can be connected to display the dynamic of water quality from land to ocean, thereby routing the import, transport, and export of pollutants. The high intensity of coastal/oceanic water quality data can be used to indicate coastal/oceanic water environment for food web (i.e., living conditions of plankton). For instance, phytoplankton and zooplankton communities are sensitive to the changes in water quality, and respond to low DO levels, high nutrient levels (i.e., DIN), and toxic contaminants (i.e., TPH). Therefore, such spatial continuous coastal/oceanic water quality dataset is helpful for characterizing the patterns of spatial-temporal distributions of plankton, assessing the status and trends of biodiversity, and predicting the population succession in the changing ocean world.

Certain studies have previously utilized specific segments of the original dataset. For instance, researchers have employed the weekly water quality data to examine the characteristics, trends, and seasonality of water quality in the Yangtze River (Di et al., 2019; Duan et al., 2018). It should be noted, however, that the complete dataset presented in this study has not been employed in any research thus far, which may limit the reliability of the dataset. In future, we plan to employ this dataset in upcoming research projects, where we will rigorously test its reliability."

We also demonstrated its contribution for application practice in Conclusion Section "Considering the extensive coverage of oceanic monitoring sites within this dataset, it has made a substantial contribution to the dissemination of coastal/oceanic water quality data, offering a comprehensive depiction of the aquatic environment, and facilitating researchers in conducting in-depth investigations into ocean ecosystem. Due to its comprehensive temporal coverage of riverine water quality data, this dataset presented a valuable adjunct for research that demands substantial datasets and continuous information."

To address the concerns of validation section you mentioned, we made substantial revisions for the data cleaning and validation process (See Section 2.2.3 below) to ensure data consistency from different data sources "

We undertook a comprehensive standardization process across all the above mentioned data providers. This harmonization encompassed the transformation of downloaded time series into a uniform file format, shifting from CSV files to R time series. Additionally, we ensured consistency in indicator selection, units, data structure, identification of missing values, and language.

Given the limited availability of indicators within the (sub)dataset, all of them were incorporated into our water quality dataset. This inclusive selection comprised both physical parameters (e.g., TEMP, TSSs) and chemical parameters (e.g., pH, BOD, COD, COD$_{Mn}$, DO, DOSAT, DIN, NH4N, NO2N, NO3N, TDP, DIP, TP, TPH, DOC, TOC). We adopted GRQA as a reference for indicator abbreviations, with the aim of facilitating international compatibility when appending to global datasets. It is noteworthy that, except for temperature (°C), pH, and DOSAT (%), the original unit of measurements for all indicators in the (sub)dataset was milligrams per liter (mg L$^{-1}$), and we

retained this unit uniformity for consistency. Eight columns (i.e., *MonitoringLocationIdentifier, LongitudeMeasure_WGS84, LatitudeMeasure_WGS84, MonitoringDate* (with the format %d/%m/%y), *IndicatorsName, Value, Unit, SourceProvider*) were then included for structuring the full dataset. Column for *MonitoringLocationIdentifier* was created as an index to connect with the metadata file.

Some observations for different indicators were merged into a single column when converting the PDF file to editable files for weekly water quality data. Those columns were selected to be divided and tidied up into several columns via regular expression automatically and validation manually. Particularly, three additional columns were added to indicate the specific year (column *MonitoringYear*), week number (column *MonitoringWeek*), and monitoring date (column *MonitoringDate*) for the weekly water quality data. The specific years and week numbers were subtracted from the filenames. The column of *MonitoringDate* for that specific week was estimated using R according to the international standard ISO 8601 that Monday was considered the first day of a week. They were validated with the descriptive text on the cover of each report that was deleted later from the weekly water quality dataset. The column of *MonitoringDate* from ocean water quality data was assumed to occur on the first day of that month to keep consistency in the date format of other datasets.

In addition, duplicated rows were identified and removed by using distinct function in R based on the unique site, indicators, monitoring week/date, and values from the (sub)datasets that included 1776 site pairs from the weekly water quality dataset due to the file inconsistencies mentioned in 2.2.1. Negative values (with 7 observations) were omitted from the weekly water quality dataset. No duplicated rows and negative values were identified from the monthly water quality datasets. In cases where 7 sites provided two daily observations but lacked specific timestamp information from the GRQA, we substituted these records with the calculated average value of the two observations. Missing (e.g., noted as '-') and empty data were replaced with NA, and were omitted from the dataset. Values that falling below known detection limits were denoted as "< DL" within the monthly water quality datasets. COD, DO, DIN, DIP, and TPH detection limits were 0.15 mg/L, 0.32 mg/L, 0.001 mg/L, 0.001 mg/L, and 0.001 mg/L, respectively. The descriptions in the stations that were originally in Chinese were replaced with Hanyu Pinyin."

(2) The title is a bit to generic for ESSD. Once the reviews are through, can you ask authors to change for a more descriptive title? (e.g. including time frames etc?)
Reply: We have specified the title "An extensive spatiotemporal water quality dataset covering four decades (1980-2022) in China."

(3) I also noticed that some of the data are in Chinese at least partially. This also needs to be fixed. Please ask for these changes during the revision phase after public discussion."
Reply: We have further clarified them with their Hanyu Pinyin.